# EchoGen: Generating Visual Echoes in Any Scene via Feed-Forward Subject-Driven Auto-Regressive Model

**Ruixiao Dong**[1,2*]  **Zhendong Wang**[1*]  **Keli Liu**[1]  **Li Li**[1†]  **Ying Chen**[2†]  **Kai Li**[2]
**Daowen Li**[2]  **Houqiang Li**[1]

[1] University of Science and Technology of China

[2] Alibaba Group

{dongruixiaoyx,zhendongwang,sa23006063}@mail.ustc.edu.cn
{lil1,lihq}@ustc.edu.cn
{chenying.ailab,kaishi.lk,lidaowen.ldw}@alibaba-inc.com

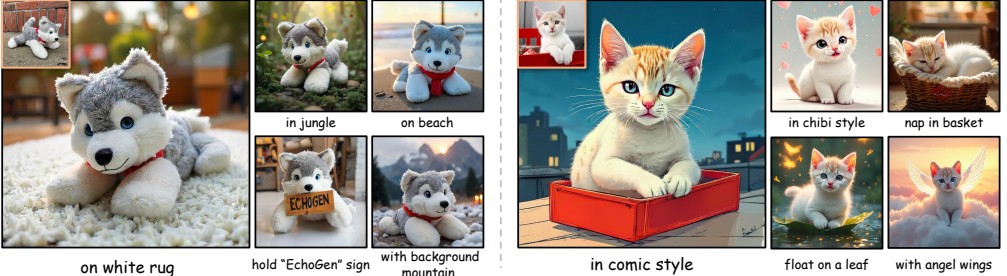

Figure 1: **Feed-forward subject-driven generation by EchoGen**. By employing a visual autoregressive paradigm, EchoGen achieves both high-quality image synthesis with lower latency, preserving intricate subject identity with exceptional efficiency.

## Abstract

Subject-driven generation is a critical task in creative AI; yet current state-of-the-art methods present a stark trade-off. They either rely on computationally expensive, per-subject fine-tuning, sacrificing efficiency and zero-shot capability, or employ feed-forward architectures built on diffusion models, which are inherently plagued by slow inference speeds. Visual Auto-Regressive (VAR) models are renowned for their rapid sampling speeds and strong generative quality, making them an ideal yet underexplored foundation for resolving this tension. To bridge this gap, we introduce **EchoGen**, a pioneering framework that empowers VAR models with subject-driven generation capabilities. The core design of EchoGen is an effective dual-path injection strategy that disentangles a subject's high-level semantic identity from its low-level fine-grained details, enabling enhanced controllability and fidelity. We employ a semantic encoder to extract the subject's abstract identity, which is injected through decoupled cross-attention to guide the overall composition. Concurrently, a content encoder captures intricate visual details, which are integrated via a multi-modal attention mechanism to ensure high-fidelity texture and structural preservation. To the best of our knowledge, EchoGen is the first feed-forward subject-driven framework built upon VAR models. Both quantitative and qualitative results substantiate our design, demonstrating that EchoGen achieves subject fidelity and image quality comparable to state-of-the-art diffusion-based methods with significantly lower sampling latency.

## 1 Introduction

The rapid evolution of text-to-image synthesis models (Saharia et al., 2022; Rombach et al., 2022; Batifol et al., 2025; Esser et al., 2024) has catalyzed a variety of novel applications (Zhang et al., 2023), among which subject-driven generation stands out as an important task. This task aims

---

[*]Equal contribution.

[†]Corresponding authors.

to accurately depict a specified subject within diverse, user-defined scenes described through text prompts, while rigorously upholding the subject's core identity. The early approaches (Ruiz et al., 2023; Gal et al., 2022; Kumari et al., 2023) introduced a test-time fine-tuning paradigm that optimizes a large pretrained model using a few images for each new subject. Although effective in preserving identity to some extent, this per-subject optimization process is computationally expensive, demanding at least hundreds of training iterations and substantial GPU resources, ultimately resulting in a distinct model checkpoint for each subject. These limitations significantly hinder the practicality and scalability of the test-time fine-tuning paradigm in real-world applications.

To improve efficiency and practicality, a new class of feed-forward approaches has recently emerged (Li et al., 2023; Pan et al., 2024; Ye et al., 2023; Tan et al., 2025; Shin et al., 2025) based on diffusion models (Rombach et al., 2022; Podell et al., 2024; Batifol et al., 2025). Instead of fine-tuning on a small set of images for each new subject, feed-forward approaches perform a single, large-scale supervised fine-tuning on a vast dataset composed of triplets (text, reference image, target image). The model is trained to learn a generalizable mapping from a subject image to the snapshot version in the specified scene. The single process of pretraining enables zero-shot generation at inference time–a novel subject can be synthesized immediately without any subject-specific fine-tuning, significantly reducing the initial setup cost and decreasing generation latency by eliminating the need for test-time optimization. Nevertheless, these methods still inherit the computational demands of the underlying diffusion models due to the iterative denoising process.

Inspired by autoregressive generation in language models (Radford et al., 2018; Achiam et al., 2023), autoregressive visual generation (Esser et al., 2021; Ramesh et al., 2021; Sun et al., 2024) has emerged as a compelling alternative to diffusion models. Unlike diffusion's iterative denoising, autoregressive models synthesize content sequentially, token by token. This paradigm is further advanced by the Visual Autoregressive (VAR) model (Tian et al., 2024; Han et al., 2025), which employs a coarse-to-fine *next-scale* generation strategy instead of traditional *next-token* generation. It first generates tokens for the global composition and then renders fine-grained details, capturing a complete hierarchical representation from structure to texture. The novel paradigm allows VAR to achieve superior performance compared to traditional autoregressive models, outperforming top-tier diffusion models while offering faster inference speed. Despite the inherent suitability of the autoregressive paradigm for fine-grained conditioning, its potential for controllable generation, especially in the feed-forward, subject-driven context, remains largely untapped compared to the wealth of research on diffusion-based methods. This critical gap severely limits the practical applicability of VAR models, hindering their adoption in real-world scenarios where subject control is paramount.

In this work, we aim to bridge this gap by leveraging the inherent advantages of VAR to build an effective, scalable, and highly controllable system for subject-driven image synthesis. We propose *EchoGen*, the first efficient **feed-forward** autoregressive framework that generates faithful visual renditions of a given subject in arbitrary scenes. At the core of EchoGen is a *dual-path* injection mechanism that disentangles semantic features from fine-grained details. We inject high-level fine-grained semantic features extracted by a semantic encoder based on the pretrained vision foundation model (DINOv2 (Oquab et al., 2024)) into the decoupled cross-attention layers (Kumari et al., 2023) to bring structural and stylistic coherence while avoiding drift in prompt following. To enable global semantic conditioning, we prepend the global semantic embedding extracted from DINOv2 as a prefix and subsequently infuse it via Adaptive LayerNorm, thereby steering the overall semantic generation. However, generating with semantic features alone often misses low-level details. To complement these features, a second pathway employs a pretrained content encoder (FLUX.1-dev VAE (Batifol et al., 2025)) to extract fine-grained image features, which are incorporated via a multi-modal attention module, ensuring faithful reconstruction of local textures and details. To preserve the generative capabilities of the pretrained VAR model, we adopt a parameter-efficient fine-tuning strategy that freezes the backbone and only updates key components within the subject injection modules. Extensive quantitative and qualitative evaluations on DreamBench (Ruiz et al., 2023) benchmark and human evaluation demonstrate that EchoGen achieves subject fidelity, text alignment, and image quality comparable to and even exceeding state-of-the-art diffusion-based methods, while exhibiting lower sampling latency.

Our principal contributions can be summarized as follows:

- We introduce EchoGen, the first feed-forward, efficient, subject-driven generation framework built upon a visual autoregressive model. This establishes a compelling new paradigm

for controllable subject-driven synthesis beyond the dominant diffusion-based approaches.

- We propose a novel dual-path injection strategy that disentangles the identity of a subject into high-level semantics and fine-grained details. By injecting these features through separate pathways within a parameter-efficiently tuned model, EchoGen achieves faithful subject representation across diverse scenes.
- Extensive experiments demonstrate that EchoGen achieves subject fidelity, text alignment, and image quality that are competitive with or superior to state-of-the-art diffusion-based methods with much faster inference speed.

## 2 RELATED WORKS

### 2.1 AUTOREGRESSIVE IMAGE GENERATION

Unlike diffusion-based methods that synthesize images via iterative denoising, the autoregressive paradigm models image distributions by sequentially predicting visual tokens conditioned on the preceding context. This approach evolves from inefficient and low quality early pixel-level methods (Van den Oord et al., 2016; Salimans et al., 2017) to a dominant two-stage framework that first compresses images into discrete tokens and then models their distribution utilizing Transformer (Esser et al., 2021). This paradigm substantially improves generation fidelity and efficiency, underpinning advances in text-to-image synthesis (Ramesh et al., 2021; Yu et al., 2022b) and controllable generation (Li et al., 2025). Subsequent work further refines it by improving image tokenizers (Yu et al., 2022a; Mentzer et al., 2024), exploring continuous representations with diffusion modeling (Li et al., 2024a; Fan et al., 2025), or adapting large language models for visual generation (Sun et al., 2024; Wu et al., 2024). To mitigate structural degradation induced by the fixed raster-scan order, Visual Autoregressive (VAR) models (Tian et al., 2024) introduce a hierarchical coarse-to-fine strategy that progressively refines fine-grained details by next-scale prediction. The following version Infinity (Han et al., 2025) extends the VAR model to text-to-image generation, achieving superior quality with significantly lower sampling latency than diffusion models. While existing works extend VAR to controllable generation (Yao et al., 2024; Li et al., 2024b; Chung et al., 2025), feed-forward subject-driven personalization remains underexplored, limiting the practical applicability of the VAR framework.

### 2.2 SUBJECT-DRIVEN IMAGE GENERATION

**Test-time fine-tuning methods**. Diffusion models (Ho et al., 2020; Rombach et al., 2022) have achieved remarkable success in high-fidelity text-to-image (T2I) synthesis (Podell et al., 2024; Esser et al., 2024; Batifol et al., 2025). For subject-driven tasks, relying solely on text prompts is often insufficient to preserve the defining characteristics of specific subjects. To address this, pioneering methods (Gal et al., 2022; Ruiz et al., 2023; Kumari et al., 2023) introduce customization by fine-tuning on a small set of reference images for each target subject. While these approaches can capture intricate details and deliver high fidelity to some extent, their dependence on per-subject optimization remains time-consuming and computationally demanding, which limits practical use.

**Feed-forward subject-driven approaches**. To overcome the efficiency limitations of per-subject optimization, feed-forward methods have been developed (Wei et al., 2023; Zeng et al., 2024; Patel et al., 2024; Ma et al., 2024; Wang et al., 2025). These models are trained once to condition on subject features from vision encoders, enabling fast, zero-shot synthesis for novel subjects. Early works such as BLIP-Diffusion (Li et al., 2023) jointly fine-tune the denoising network with multimodal alignment modules while suffering from inadequate fidelity and image quality. To mitigate the high computational cost of full model tuning, parameter-efficient strategies (Pan et al., 2024; Ye et al., 2023; Tan et al., 2025; Zhang et al., 2025; Wu et al., 2025) incorporate lightweight modules such as LoRA (Hu et al., 2022) or adapters. These modules inject reference features into the diffusion transformer, typically via attention mechanisms, while keeping most pretrained weights frozen. However, since these methods all rely on diffusion backbones, they inherit the substantial inference latency of the iterative denoising process, which constrains their practical deployment.

## 3 PRELIMINARY OF VISUAL AUTOREGRESSIVE MODELING

Autoregressive models (Esser et al., 2021) reframe image synthesis as a sequential token prediction, under the *next-token* prediction. The image is first tokenized into a discrete feature map us-

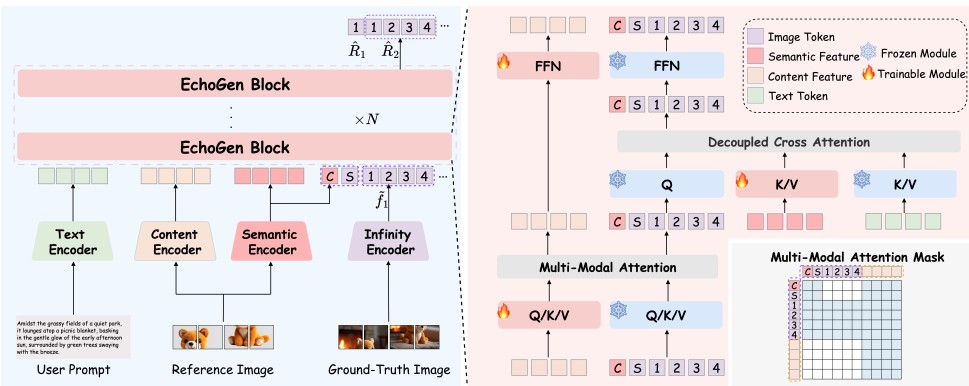

Figure 2: **Overview of the EchoGen architecture**. The left panel illustrates the overall model framework with dual-path subject injection, while the right panel provides a detailed schematic of the EchoGen block with a carefully designed attention mask applied in the Multi-Modal Attention module to avoid feature leakage. $C$ denotes the global semantic token extracted from the semantic encoder, which is prepended to the input sequence. $S$ represents the start token for the first-scale generation. Adaptive Layer Normalization modules in the EchoGen blocks are omitted for clarity.

ing a visual tokenizer $\mathcal{E}$ and then flattened into a one-dimensional sequence, typically following the raster scan order. The model is then trained to predict each token $x_i$ given the preceding tokens $(x_1, ..., x_{i-1})$ and the condition $c$, factorizing the sequence distribution as $p(x_1, \ldots, x_N|c) = \prod_{i=1}^{N} p(x_i|x_1, \ldots, x_{i-1}, c)$. However, the vanilla next-token paradigm with fixed raster order induces structural degradation and insufficient modeling.

Visual autoregressive modeling (Tian et al., 2024) addresses the above issues by shifting the prediction paradigm from *next-token* to *next-scale*: instead of predicting one token at a time, it predicts entire token maps at *progressively increasing resolutions*. The visual encoder $\mathcal{E}$ first maps an image $I$ to latent $F$, and then produces $K$ multi-scale token maps $(r_1, ...r_K)$ with increasing resolutions $h_k \times w_k$ by applying a residual vector quantizer. A GPT-style Transformer begins from the generation of the $1 \times 1$ map $r_1$ and autoregressively predicts each subsequent scale given prior scales and condition $c$, achieving generation from global structure to fine details, which is formulated as:

$$p(r_1, \ldots, r_K|c) = \prod_{k=1}^{K} p(r_k|r_1, \ldots, r_{k-1}, c). \tag{1}$$

This scale-wise coarse-to-fine paradigm is well suited for scalable text-to-image generation. The text-to-image generation model Infinity (Han et al., 2025) leverages bitwise quantization to expand the vocabulary size under the next-scale paradigm, reporting state-of-the-art performance with reduced sampling latency compared to diffusion baselines. In this paper, to bypass the cumbersome per-subject fine-tuning and the heavy computational cost during inference, we propose a novel feed-forward framework based on VAR models, featuring a single parameter-efficient fine-tuning phase.

## 4 ECHOGEN

### 4.1 OVERALL FRAMEWORK

We are seeking a novel feed-forward framework for subject-driven generation built upon Infinity, based on the proposed **EchoGen** block with effective dual-path subject information injection, in which a content encoder and a semantic encoder cooperate to provide comprehensive subject features from both sides of a coin. The overview of the EchoGen architecture and its basic block is illustrated in Figure 2. Before subject injection, to ensure robustness against background noise that may interfere with subject injection, a pipeline based on the multi-modality model Qwen2.5-VL (Bai et al., 2025) and the open segmentation model GroundingDINO (Liu et al., 2024) is carefully designed to segment the subject from complex scenes. Given the segmented subject image, our EchoGen model is trained using a parameter-efficient methodology that freezes the pretrained backbone while fine-tuning only newly introduced attention modules. During inference, we apply

flexible subject-text classifier-free guidance for explicit control over the trade-off between subject fidelity and textual alignment, enabling versatile and controllable generation.

## 4.2 DUAL-PATH SUBJECT INJECTION

**Semantic feature injection for identity preservation**. The semantic feature, which captures abstract characteristics, provides a representation that is critical for avoiding the identity drift common in subject-driven generative models. Following this principle, we introduce a bifurcated injection strategy that targets both the fine-grained and global levels of the generative process. For fine-grained conditioning, we employ the pretrained DINOv2 vision encoder to extract patch-level semantic embeddings. These embeddings are synergistically integrated with the original textual conditioning via a decoupled cross-attention mechanism (Kumari et al., 2023). Our decoupled cross-attention mechanism operates on query features $\mathcal{Z}$, conditioning them on both the text embedding $c_t$ and the fine-grained semantic features $c_s$, formulated as follows:

$$\mathcal{Q} = \mathcal{Z} W^q, \ \mathcal{K} = \text{concat}\left(c_s W_s^k, c_t W_t^k\right), \ \mathcal{V} = \text{concat}\left(c_s W_s^v, c_t W_t^v\right),$$
$$\mathcal{Z}' = \text{Attention}\left(\mathcal{Q}, \mathcal{K}, \mathcal{V}\right) = \text{Softmax}\left(\mathcal{Q}\mathcal{K}^\top / \sqrt{d}\right)\mathcal{V}, \tag{2}$$

where $W^q$ is the query projector, $(W_t^k, W_t^v)$ and $(W_s^k, W_s^v)$ are two distinct sets of $(k, v)$ projectors to embed text prompting $c_t$ and semantic injection $c_s$, respectively. The resulting key and value pairs for each condition are concatenated to form the final context vectors $\mathcal{K}$ and $\mathcal{V}$. We keep the projectors for text prompting $(W_t^k, W_t^v)$ and the query projector $W^q$ frozen while exclusively optimizing the key and value projectors $(W_s^k, W_s^v)$ that map the semantic features of the reference images, enabling an alignment mapping from the semantic visual space to the generator's latent space without perturbing the pretrained knowledge.

Moreover, we prepend the DINOv2 global semantic token $C$ to the input sequence to impose holistic semantic guidance. At the same time, this global token also serves as a condition for the Adaptive Layer Normalization (AdaLN) layer in the proposed EchoGen block, following (Han et al., 2025). The infusion of fine-grained and global semantics ensures comprehensive semantic-informed generation, promoting fine-grained fidelity and global structural coherence.

**Content feature injection for detail preservation**. While the semantic embeddings provide a robust identity preservation, their high abstraction leads to generation with insufficient subject details. To achieve high fidelity of the subject's content, we complement it with a content feature infusion mechanism. To be specific, EchoGen employs the FLUX.1-dev VAE to extract low-level content features $c_c$, which are then integrated via the multi-modal attention. The generation process is then steered by a carefully designed attention operation: generated tokens have unobstructed access to the reference tokens, allowing them to distill fine-grained visual cues on demand; conversely, a causal mask renders the reference tokens oblivious to the generated sequence, which is a critical constraint for ensuring the autoregressive sampling trajectory. This masking schema is precisely demonstrated in the lower-right inset of Figure 2. Specifically, given the generated token sequence $\mathcal{Z}$ and the detailed content condition $c_c$, the multi-modal attention utilizes separate linear projections $(W^q, W^k, W^v)$ for $\mathcal{Z}$ and $(W_c^q, W_c^k, W_c^v)$ for the condition $c_c$, with the applied attention mask Mask, and then calculate the generated sequence $\mathcal{Z}'$ and condition $c_c'$ via:

$$\mathcal{Q} = \text{concat}\left(\mathcal{Z} W^q, c_c W_c^q\right), \ \mathcal{K} = \text{concat}\left(\mathcal{Z} W^k, c_c W_c^k\right), \ \mathcal{V} = \text{concat}\left(\mathcal{Z} W^v, c_c W_c^v\right),$$
$$\mathcal{Z}', c_c' = \text{Attention}\left(\mathcal{Q}, \mathcal{K}, \mathcal{V}, \text{Mask}\right) = \text{Softmax}\left(\text{Mask}\left(\mathcal{Q}\mathcal{K}^\top / \sqrt{d}\right)\right)\mathcal{V}. \tag{3}$$

The pathways for the generated token sequence remain frozen, while exclusively parallel attention projectors $(W_c^q, W_c^k, W_c^v)$ and FFN modules for processing content features are optimized.

Through this dual-path subject injection strategy, our model faithfully preserves the salient visual characteristics of the reference image while simultaneously maintaining a strong adherence to the provided text instructions.

## 4.3 SUBJECT SEGMENTATION

A common challenge in real-world scenarios is that user-provided reference images comprised of

the subject of interest within visually complex backgrounds may harm the performance of subject injection. To mitigate this issue, we employ a subject segmentation pre-processing pipeline, illustrated in Figure 3. First, the Qwen2.5-VL (Bai et al., 2025) vision-language model

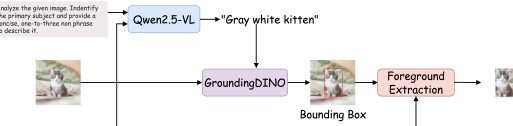

Figure 3: **The pipeline of subject segmentation**.

identifies the subject's semantic identity, producing a descriptive text prompt. This prompt is then used to condition the GroundingDINO (Liu et al., 2024) model for precise subject localization and bounding box generation. The foreground region is subsequently cropped according to this bounding box, while the surrounding unrelated regions are explicitly discarded and replaced with a uniform white background. This process ensures that subsequent feature injection operates attentively on the isolated representation of the referred subject.

## 4.4 Sampling with Subject-Text Classifier-free Guidance

Classifier-Free Guidance (CFG) (Ho & Salimans, 2021) has become a cornerstone technique for enhancing conditional control in generative models, especially in diffusion models. Its core principle is to amplify the conditional signal by extrapolating from an unconditional prediction towards a conditional one, thereby improving condition following at the cost of some diversity. Recently, many autoregressive models (Chang et al., 2023; Tian et al., 2024) have also incorporated CFG into their frameworks. In this work, we further enhance the influence of the text embedding $c_t$ and the subject condition $c_s, c_c$ within the CFG scheme for subject-driven generation. During training, we independently replace the text condition $c_t$ with an unconditional token $\varnothing_t$ and the image condition $c_s, c_c$ with unconditional embeddings $\varnothing_s, \varnothing_c$, each with a probability of 10%. During inference, assuming the independence between the text condition $c_t$ and the image condition $c_s, c_c$, we compute the final logits predicted by EchoGen via a flexible guidance rule that integrates both controls:

$$\hat{l} = l(\varnothing_t, \varnothing_s, \varnothing_c) + \gamma_t \times (l(c_t, \varnothing_s, \varnothing_c) - l(\varnothing_t, \varnothing_s, \varnothing_c)) + \gamma_I \times (l(c_t, c_s, c_c) - l(c_t, \varnothing_s, \varnothing_c)), \quad (4)$$

where $l$ denotes the Transformer output logits, and $\gamma_t$ together with $\gamma_I$ are hyperparameters that govern the guidance scales. This dynamic text-subject guidance not only strengthens the influence of text embeddings and image prompts, thereby improving generation performance, but also provides a flexible mechanism to balance text alignment with the reference preservation.

## 5 Experiment

### 5.1 Setup

**Datasets**. We conduct experiments on a merged dataset curated from the Subjects200K (Tan et al., 2025) and UNO-1M datasets (Wu et al., 2025), yielding a large-scale high-quality corpus of approximately 640,000 triplets (text prompts, reference images, and target images). The corpora were synthetically generated using large language models (*e.g.*, GPT-4o) and text-to-image generative models (*e.g.*, FLUX.1-dev), and with image resolutions larger than 500×500. For EchoGen-0.1B training, both the reference and target images are resized and center-cropped to 256×256. To enable high-resolution generation for EchoGen-2B training, we avoid direct interpolation, which may introduce undesirable artifacts; instead, we upscale the images to 1024×1024 using the PiSA-SR super-resolution model (Sun et al., 2025).

**Training details**. Our training protocol largely follows Infinity (Han et al., 2025). We train EchoGen for 80K iterations, utilizing the AdamW (Loshchilov & Hutter, 2017) optimizer with a global batch size of 128, setting the base learning rate as $3 \times 10^{-5}$ and the momentum parameters $(\beta_1, \beta_2) = (0.9, 0.97)$. To stabilize fine-tuning, we apply a reduced learning rate of $3 \times 10^{-6}$ to the multi-modal attention parameters. More training details can be found in the appendix 7.3.1.

**Evaluation**. Following prior works (Ruiz et al., 2023; Li et al., 2023), we evaluate our approach in terms of subject fidelity and text alignment on the DreamBench benchmark (Ruiz et al., 2023). Subject fidelity is measured by the cosine similarity between the generated and reference images using both CLIP (Radford et al., 2021) image embeddings (CLIP-I) and DINO (Zhang et al., 2022) features (DINO). Text alignment is assessed via the CLIP cosine similarity between the generated image and its corresponding input prompt (CLIP-T). DreamBench, comprising real-world images with prompt

| Method | Base Model | DINO↑ | CLIP-I↑ | CLIP-T↑ | Latency ↓ |
|---|---|---|---|---|---|
| *Test-time Fine-tuning* | | | | | |
| Textual-Inversion (Gal et al., 2022) | SD-v1.5 | 0.569 | 0.780 | 0.255 | 50min |
| DreamBooth (Ruiz et al., 2023) | SD-v1.5 | 0.668 | 0.803 | 0.305 | 15min |
| BLIP-Diffusion (Li et al., 2023) | SD-v1.5 | 0.670 | 0.805 | 0.302 | - |
| AR-Booth (Chung et al., 2025) | Infinity-2B | 0.750 | 0.808 | 0.269 | 2.8h |
| *Unified Generation* | | | | | |
| OmniGen (Xiao et al., 2025) | OmniGen | 0.693 | 0.801 | 0.315 | 93.4s |
| *Feed-Forward* | | | | | |
| ELITE (Wei et al., 2023) | SD-v1.4 | 0.621 | 0.771 | 0.293 | 11.0s |
| Re-Imagen (Chen et al., 2023) | Imagen | 0.600 | 0.740 | 0.270 | - |
| BLIP-Diffusion (Li et al., 2023) | SD-v1.5 | 0.594 | 0.779 | 0.300 | - |
| $\lambda$-Eclipse (Patel et al., 2024) | Kan-v2.2 | 0.613 | 0.783 | 0.307 | - |
| MS-Diffusion (Wang et al., 2025) | SDXL | 0.671 | 0.792 | 0.321 | 39.6s |
| IP-Adapter (Ye et al., 2023) | SDXL | 0.613 | 0.810 | 0.292 | 16.9s |
| IP-Adapter (Ye et al., 2023) | FLUX.1-dev | 0.561 | 0.725 | 0.351 | - |
| OminiControl (Tan et al., 2025) | FLUX.1-dev | 0.684 | 0.799 | 0.312 | 27.5s |
| EasyControl (Zhang et al., 2025) | FLUX.1-dev | 0.652 | 0.789 | 0.325 | 25.4s |
| EchoGen-0.1B | Infinity-0.1B | 0.675 | 0.806 | 0.321 | 0.5s |
| EchoGen-2B | Infinity-2B | 0.755 | 0.835 | 0.325 | 5.2s |

Table 1: **Quantitative comparisons on DreamBench (Ruiz et al., 2023)**. We highlight the best , second-best , and third-best values for each metric. The results indicate that EchoGen attains performance on par with diffusion-based approaches while delivering substantially faster sampling.

| Method | Subject Fidelity↑ | Text Alignment↑ | Photorealism↑ |
|---|---|---|---|
| OmniGen (Xiao et al., 2025) | 0.15 | 0.13 | 0.09 |
| IP-adapter (Ye et al., 2023) | 0.21 | 0.05 | 0.14 |
| OminiControl (Tan et al., 2025) | 0.12 | 0.21 | 0.15 |
| EasyControl (Zhang et al., 2025) | 0.15 | **0.31** | 0.28 |
| EchoGen-2B | **0.37** | 0.30 | **0.34** |

Table 2: **Human evaluation**. We compare our method with previous approaches based on three aspects: text alignment, subject fidelity, and photorealism.

annotations, includes 30 unique subjects, each paired with 25 distinct prompts. Following the evaluation protocol of (Pan et al., 2024), we select one reference image per subject, generate four images for each prompt–subject pair, yielding 3,000 generated images in total. DINO and CLIP-I scores are computed by comparing each generated image against its corresponding reference image. The sampling latency is measured on an H20 GPU for all methods.

## 5.2 MAIN RESULTS

We compare EchoGen with three categories of prior works: (1) test-time fine-tuning methods that require per-subject optimization; (2) unified generation models with large-scale pre-training; and (3) feed-forward approaches that share the same paradigm as ours and constitute our most baselines.

**Quantitative results**. We benchmark EchoGen performance against contemporary subject-driven diffusion-based methods in the DreamBench dataset (Ruiz et al., 2023), with quantitative results summarized in Table 1. EchoGen achieves performance that comparable or superior to leading diffusion-based approaches in the core metrics of subject fidelity and text alignment, and demonstrates balanced performance across evaluation axes. In contrast, several baselines, such as IP-Adapter (Ye et al., 2023) exhibit significant weaknesses in specific metrics. Furthermore, the adoption of the visual autoregressive paradigm provides a clear efficiency advantage: EchoGen's inference latency for a 1024×1024 image is under 6 seconds, representing a significant acceleration over the more than 10 seconds required by its diffusion-based counterparts. Overall, these results indicate that EchoGen combines strong generative quality with markedly improved efficiency, offering

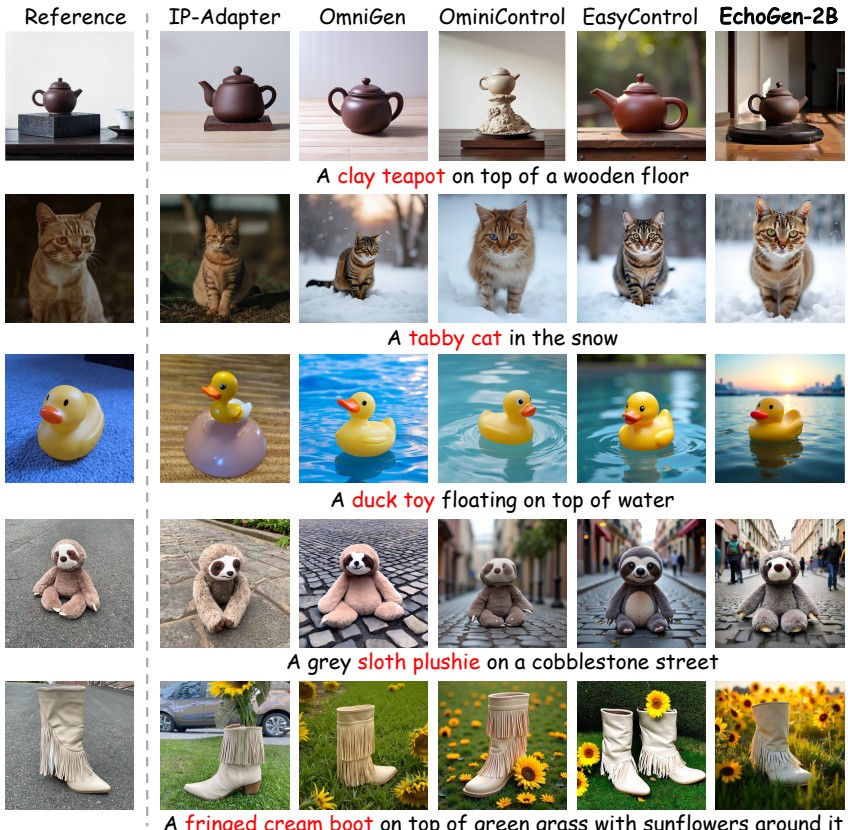

Reference    IP-Adapter    OmniGen    OminiControl    EasyControl    **EchoGen-2B**

A clay teapot on top of a wooden floor

A tabby cat in the snow

A duck toy floating on top of water

A grey sloth plushie on a cobblestone street

A fringed cream boot on top of green grass with sunflowers around it

Figure 4: **Qualitative comparison with diffusion-based methods on DreamBench (Ruiz et al., 2023)**. For a fair comparison, we adopt the default sampling settings for all baseline models.

a competitive alternative for subject-driven synthesis. Since existing methods often adopt different evaluation protocols, we re-implement representative diffusion-based baselines and compare them with our model under a unified setting (see Section 7.4). The results demonstrate that our approach achieves comparable or superior performance while consistently reducing inference latency.

**Qualitative results**. Figure 4 presents a rigorous qualitative comparison with prominent diffusion-based frameworks, revealing substantial advantages of our model in both subject fidelity and prompt correspondence. EchoGen exhibits the ability to render high-fidelity details, such as the precise reconstruction of the teapot spout and the nuanced texture of the sloth plushie, and we attribute this capability to our dual-path semantic-content feature injection design. In contrast, baselines including IP-Adapter (Ye et al., 2023) and OmniControl (Tan et al., 2025) exhibit characteristic failure cases, corroborating EchoGen's robustness. EchoGen also demonstrates more consistent compliance with textual prompts, avoiding the language deviations observed in the generations of the duck toy and cat instances by IP-Adapter.

**Human evaluation**. To assess the perceptual quality of EchoGen, we conduct a human evaluation study against strong baselines that span multiple categories of subject-driven methods. We focus on three criteria: text alignment, subject fidelity, and photorealism. The images are generated conditioned on the reference images and prompts sampled from DreamBench (Ruiz et al., 2023) and DreamBench++ (Peng et al., 2024) benchmarks without cherry-picking, and for each criterion, participants select their preferred generated image among the outputs from five methods. We collect 450 responses from 25 participants, all with expertise in generative models, and report preference ratios in Table 2. The results show that EchoGen is preferred for subject fidelity and photorealism, surpassing all the diffusion-based contemporary baselines on these criteria. For text alignment, EchoGen performs on par with EasyControl (Zhang et al., 2025) and exhibits a clear advantage over the other compared methods.

**Sampling Latency Analysis.** We conduct a thorough analysis to evaluate the performance-latency trade-offs across all methods. Specifically, we re-produce diffusion-based methods with varying numbers of denoising steps in our evaluation protocol and report their performance versus sampling

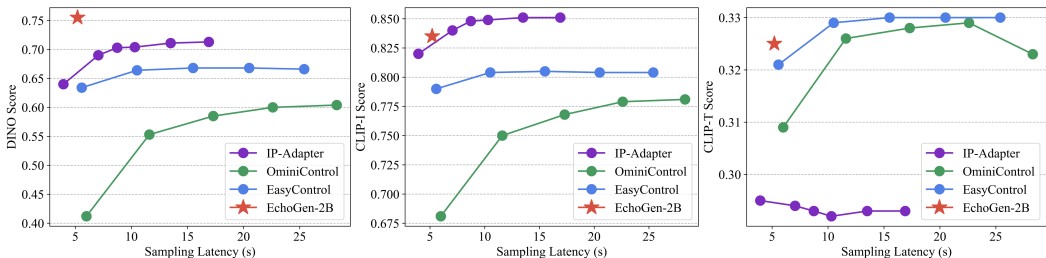

Figure 5: **Performance *v.s.* sampling latency comparison among our EchoGen and baselines.**

| Enc. | DINO↑ | CLIP-I↑ | CLIP-T↑ |
|------|-------|---------|---------|
| SigLIP-2 | 0.438 | 0.720 | 0.320 |
| FLUX.1-dev | 0.433 | 0.706 | 0.320 |
| DINOv2 | **0.632** | **0.788** | **0.328** |

Table 4: **Significance of fine-grained semantic injection**. "Enc." denotes the encoder type.

| Exp. | DINO↑ | CLIP-I↑ | CLIP-T↑ |
|------|-------|---------|---------|
| w/o prefix | 0.632 | 0.788 | **0.328** |
| w prefix | **0.670** | **0.798** | 0.322 |

Table 5: **Ablation study on incorporating the global semantic features of reference images**.

latency in Figure 5. For the diffusion baselines, increasing the number of denoising steps improves subject fidelity (as measured by DINO, CLIP-I scores) up to a saturation point. In contrast, the text alignment (CLIP-T) score converges much earlier. Our model consistently offers a better trade-off, achieving comparable performance with significantly lower sampling latency than the diffusion-based baselines. This confirms the inherent efficiency and effectiveness of our approach.

The detailed component-wise sampling latency of our EchoGen framework is provided in Table 3. The results confirm that the framework's overall efficiency is not limited by auxiliary components such as Grounding-DINO. EchoGen maintains a significant speed advantage over diffusion-based methods, even with the inclusion of the optional Qwen2.5-VL model. Although this model is employed during training to automate subject identification for the GroundingDINO segmentation model, it is not required during inference. Instead, users can provide a descriptive text prompt (akin to the DreamBench format) for specifying the subject.

| Model Component | Sampling Latency (s) |
|-----------------|---------------------|
| Grounding-DINO | 0.24 |
| Semantic encoder | 0.01 |
| Content encoder | 0.02 |
| Infinity generator | 4.95 |
| Qwen2.5-VL(Optional) | 1.13 |
| EchoGen (w/o Qwen2.5-VL) | 5.22 |
| EchoGen (w/ Qwen2.5-VL) | 6.35 |

Table 3: **Per-component sampling latency measured on a single H20 GPU.**

## 5.3 ABLATION STUDIES

We conduct a series of ablation studies to verify the effect of each component in EchoGen. Owing to computational constraints, all experiments are conducted on EchoGen-0.1B, using the same training settings to ensure fair ablation studies.

**Significance of fine-grained semantic information injection**. Fine-grained semantic conditional information is critical as it provides guidance for establishing the structure, enabling the model to synthesize stylistically and structurally coherent features consistent with the subject. Conversely, we argue that overly coarse-grained semantic features may fail to provide sufficient guidance for generating visually consistent echoes. To validate the importance of incorporating fine-grained semantic information, we conducted an ablation study with three distinct feature types independently injected via cross-attention:(1) coarse-grained semantic identity from SigLIP-2 (Tschannen et al., 2025), (2) fine-grained semantic features from DINOv2 and (3) FLUX.1-dev VAE features, which lack enough semantic information. Table 4 demonstrates that the fine-grained semantic DINOv2 features are the most suitable to represent the echo information in this task, as evidenced by all criteria. The failure of the SigLIP-2 and FLUX.1-dev VAE features can be attributed to their respective limitations: the former relies on features that are too coarse to guide subject generation, while the latter lacks semantic information.

**Incorporating the global semantic information**. To provide stronger guidance when constructing the global structure in the generation process, we prefix the global semantic $C$ token extracted by the

DINOv2 semantic encoder and incorporate this token into the model via Adaptive LayerNorm. As demonstrated in Table 5, compared with solely relying on injecting fine-grained semantic features via cross-attention, the introduction of the global semantic token yields a substantial performance gain in consistency of subject characteristics, validating the effectiveness of incorporating global semantic information.

**Distinct semantic feature injection strategies**. We explore the most effective method to guide the synthesis process conditioned on the semantic features of reference images. Table 6 presents an analysis comparing two distinct feature injection modules: multi-modal attention and cross-attention. Our results indicate

| Module | DINO↑ | CLIP-I↑ | CLIP-T↑ |
|--------|-------|---------|---------|
| MM-Attn | 0.646 | 0.792 | **0.325** |
| Cross-Attn | **0.670** | **0.798** | 0.322 |

Table 6: **Different methods for incorporating semantic features**.

that while the multi-modal module achieves slightly better alignment with text prompts, the cross-attention mechanism yields significantly superior subject fidelity, as evidenced by a notably higher DINO score. Based on this, we opted to utilize cross-attention for injecting the semantic features in all subsequent experiments, rather than the multi-modal attention.

**Enhancing subject fidelity with detailed content features.** Considering the absence of local details in the semantic features of the reference images, we incorporate a secondary pathway that injects localized content features of the subject. These features, extracted by the FLUX.1-dev VAE, are used to guide the syn-

| Exp. | DINO↑ | CLIP-I↑ | CLIP-T↑ |
|------|-------|---------|---------|
| baseline | 0.670 | 0.798 | **0.322** |
| +Cross-Attn | 0.667 | 0.803 | 0.318 |
| +MM-Attn | **0.672** | **0.806** | 0.321 |

Table 7: **Impact of injecting subject details.**

thesis of the fine-grained local details of the subject. The ablation study detailed in Table 7 shows that employing a multi-modal attention mechanism to infuse these content features substantially improves the subject-fidelity of the generated samples, yielding a significant increase in CLIP-I.

**Qualitative analysis of the effect of semantic and content feature injection.** We further qualitatively dissect the effect of each feature component to validate our design. As shown in Figure 6, starting from the base Infinity backbone, introducing semantic features extracted by DINOv2 enables the generator to synthesize subjects that faithfully preserve the reference subject's structure and style. Moreover, further incorporating content features from the FLUX.1-dev VAE significantly enhances EchoGen's capability to render fine-grained, coherent details (*e.g.*, the facial features of the robot toy and the fluffy dog, as well as the material and color of the shoe uppers). These qualitative re-

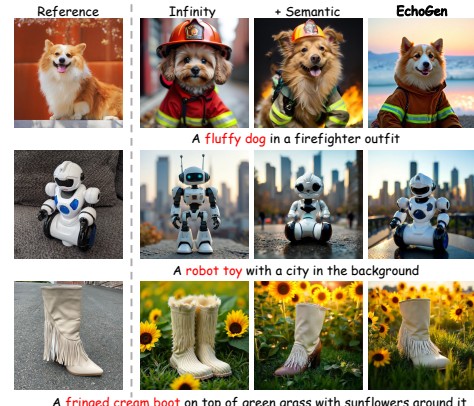

Figure 6: **Qualitative analysis of the effect of semantic and content feature injection.**

sults confirm the effectiveness of our dual-path injection design, where semantic and content features play distinct yet complementary roles.

# 6 CONCLUSION

This paper presents EchoGen, a novel framework for efficient, feed-forward subject-driven image synthesis based on a visual autoregressive paradigm, aiming to inherit the properties of high-quality generation and fast inference speed. Central to our methodology is a dual-path injection mechanism, meticulously designed to integrate both the semantic attributes and the precise textural details of reference images. Comprehensive evaluations corroborate the superiority of our design, revealing that EchoGen achieves generative performance on par with leading diffusion models while exhibiting substantially lower sampling latency. By pioneering a feed-forward, autoregressive solution for subject-driven synthesis, this research charts a new trajectory for the future development and application of visual autoregressive generative models.

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

# 7 APPENDIX

## 7.1 DATASET

We train EchoGen on the filtered combination of the Subjects200K and UNO-1M synthetic datasets. The Subjects200K dataset[1], introduced by (Tan et al., 2025), contains approximately 256,000 triplets, each comprising a reference image, a text prompt, and a corresponding generated target image. It is specifically established for subject-driven generation. The dataset is constructed by first using GPT-4o to produce over 30,000 diverse subject descriptions that each description represents the same subject across multiple scenes. These descriptions are then reformulated into structured prompts, each specifying a single subject in two different scenes, which are fed into FLUX.1-dev text-to-image model to synthesize paired images. Finally, GPT-4o filters the generated pairs to ensure subject consistency and overall image quality. All images have the resolution above 500 pixels, providing sufficient detail for training.

The UNO-1M dataset[2] (Wu et al., 2025) comprises approximately 1M data triplets and is built in a similar manner. It leverages the LLM to generate diverse subject instances and scenes guided by a taxonomy tree derived from the 365 general classes of Object-365, and then employs FLUX.1 model to synthesize image pairs. After that, DINO-v2 and Vision–Language Models (VLMs) are further used to score and filter the generated data. To obtain a high-quality corpus, we use their single-object subset and additionally filter it based on the VLM scores, resulting in approximately 394,000 triplets.

## 7.2 PSEUDO-CODE OF THE ECHOGEN BLOCK

We provide a PyTorch-style pseudocode for our EchoGen Block in Algorithm 1 to facilitate reproducibility and clarity.

## 7.3 IMPLEMENTATION DETAILS

### 7.3.1 TRAINING DETAILS

**Data Preprocssing**. All training images undergo a standardized pre-processing pipeline. Initially, images are resized so that their shorter edge matches the model's target resolution, which is 256 pixels for EchoGen-0.1B and 1024 pixels for EchoGen-2B, followed by a central crop to achieve a square aspect ratio.

To maintain data quality for high-resolution image generation within the EchoGen-2B model, we circumvent the quality degradation induced by naive bilinear upsampling. Instead of using simple bilinear scaling on images smaller than 1024 pixels, we integrate a super-resolution step. Specifically, we leverage the PiSA-SR model (Sun et al., 2025) to upscale these images, a method chosen to preserve fine-grained textures and prevent the introduction of common interpolation artifacts. This ensures that the model is trained exclusively on high-quality and high-resolution exemplars.

**Training Hyper-parameters**. We follow the Infinity-2B standard training recipe (Han et al., 2025), and the detailed hyperparameter configurations used to train our EchoGen are provided in Table 8. To mitigate error accumulation, as mentioned in Infinity, we employ bitwise self-correction by randomly flipping bits in the input sequence with a probability of 0.3. To improve robustness against variations in instruction length, prompts are randomly truncated to a single sentence with a probability of 0.5 during training. The EchoGen models are trained using 32 H20 GPUs, requiring 2 weeks for the longest schedule (training our EchoGen-2B model for 20 epochs).

### 7.3.2 EVALUATION DETAILS

For our quantitative evaluation, we utilize the DreamBench dataset (Ruiz et al., 2023). The dataset comprises 30 distinct subjects, categorized into 9 animate pets (cats and dogs) and 21 diverse inanimate objects (*e.g.*, toys, sunglasses, backpacks). Each subject is associated with 25 textual prompts specifically designed to test the model's abilities in recontextualization, property modification, and

---

[1]https://huggingface.co/datasets/Yuanshi/Subjects200K
[2]https://huggingface.co/datasets/bytedance-research/UNO-1M

```
class EchoGenBlock(nn.Module)
  def __init__(dim, mask):
    # Multi-modal attention QKV projectors for the image token sequence
    self.qkv_mm = nn.Linear(dim, 3*dim)
    # Multi-modal attention QKV projectors for the detailed content feature
    self.qkv_mm_c = nn.Linear(dim, 3*dim)
    self.mask = mask

    # Cross attention query projectors for the image token sequence
    self.q_ca = nn.Linear(dim, dim)
    # Cross attention KV projectors for the semantic feature
    self.kv_ca_s = nn.Linear(dim, 2*dim)
    # Cross attention KV projectors for the text embedding
    self.kv_ca_t = nn.Linear(dim, 2*dim)

    # FFN for the image token sequence
    self.ffn = MLP(dim)
    # FFN for the detailed content feature
    self.ffn_c = MLP(dim)

  def forward(self, x, cc, cs, ct):
    # Multi-modal attention
    q, k, v = self.qkv_mm(x)
    qc, kc, vc = self.qkv_mm(cc)

    q = torch.concat((q, qc))
    k = torch.concat((k, kc))
    v = torch.concat((v, vc))
    x, cc = attention(q, k, v, self.mask)

    # Cross attention
    q = self.q_ca(x)
    ks, vs = self.kv_ca_s(cs)
    kt, vt = self.kv_ca_t(ct)

    k = torch.concat((ks, kt))
    v = torch.concat((vs, vt))
    x = attention(q, k, v, mask=None)

    # Feed-forward network
    x = self.ffn(x)
    cc = self.ffn_c(cc)

    return x, cc, cs, ct
```

**Pseudo-code illustrating the EchoGen Block**. Here, $x$ denotes the image token sequence, and the generation process is conditioned on the semantic feature $c_s$ and detailed content feature $c_c$ extracted from the reference image, along with the text embedding $c_t$.

accessorization. Our data preparation protocol is adapted from (Pan et al., 2024), which involves selecting a single reference image per subject and augmenting its subject identity phrase with descriptive keywords. The correspondence between the DreamBench dataset directory name and the augmented subject description is summarized as follows:

- backpack, backpack
- backpack_dog, dog shaped backpack
- bear_plushie, bear plushie
- can, 'Transatlantic IPA' can
- candle, jar candle
- cat, tabby cat
- cat2, grey cat
- clock, number '3' clock
- colorful_sneaker, colorful sneaker
- dog1, fluffy dog
- dog2, fluffy dog
- dog3, curly-haired dog

| Config | value |
|---|---|
| Bitwise Self-correction Flip Ratio | 0.3 |
| Bitwise Self-correction Apply Layers | 13 |
| Dynamic Truncate Prompt Ratio | 0.5 |
| Infinity Image Encoder Channel | 16(0.1B) / 32(2B) |
| Text Encoder | Flan-t5-xl |
| Text Embedding Channels | 2048 |
| Maximum Text Tokens Length | 512 |
| Semantic Image Encoder | DINO-v2-Base |
| Semantic Feature Channels | 768 |
| Semantic Downsample ratio | 14 |
| Content Image Encoder | FLUX.1-dev VAE |
| Content Feature Channels | 16 |
| Content Downsample ratio | 8 |
| Reweight Loss by Scale | True |
| Gradient clipping by norm | 5.0 |
| Optimizer | Adamw |
| Beta1 | 0.9 |
| Beta2 | 0.97 |
| Decay | 0 |
| Base Learning rate | 3e-5 |
| Multi-Modal Modules Learning rate | 3e-6 |
| Learning rate warmup iterations | 0 |
| Training epochs | 20 |
| Total Batchsize | 128 |
| GPU | H20 |

Table 8: **Detailed hyper-parameters for training our EchoGen**.

- dog5, long-haired dog
- dog6, puppy
- dog7, dog
- dog8, dog
- duck_toy, duck toy
- fancy_boot, fringed cream boot
- grey_sloth_plushie, grey sloth plushie
- monster_toy, monster toy
- pink_sunglasses, sunglasses
- poop_emoji, poop-emoji shaped toy
- rc_car, car toy
- red_cartoon, cartoon character
- robot_toy, robot toy
- shiny_sneaker, sneaker
- teapot, clay teapot
- vase, tall vase
- wolf_plushie, wolf plushie

We compute DINO and CLIP-I scores by comparing each generated image with its corresponding single reference image in our main experiments and ablation studies. Note that some methods (Li

et al., 2023) especially test-time fine-tuning methods (Ruiz et al., 2023) instead compute DINO and CLIP-I by comparing a generated image against all images of the same entity in DreamBench. To enable a more fair comparison under both protocols, we evaluate EchoGen and several leading baselines using unified implementation and report all results in Section 7.4.

To augment the diversity and rigor of our human evaluation, we incorporate a curated set of instances from the DreamBench++ benchmark. DreamBench++ includes 150 subjects, each paired with nine prompts.

## 7.4 MORE ABLATION STUDIES

In this section, we present additional ablation studies to analyze the individual components of EchoGen. These ablation studies are also conducted based on EchoGen-0.1B model with fair training settings.

**Importance of injecting global semantic information**. Injecting global semantic information serves as a prepended condition, to ensure global structural coherence during generation. Our ablation study in Table 5 confirms the significant benefits of incorporating global semantic features. Moreover, we conduct a targeted experiment comparing the injection of global semantic versus content features to the Image Token. As shown in Table 9, the results clearly indicate that prepending semantic features rather than content information into the image token significantly enhances the subject fidelity. This confirms that our choice to inject global semantic guidance into image tokens is both effective and well-justified.

| Exp. | DINO↑ | CLIP-I↑ | CLIP-T↑ |
|---|---|---|---|
| Content | 0.663 | 0.795 | **0.322** |
| Semantic | **0.672** | **0.806** | 0.321 |

Table 9: **Analysis of different information types prepended to the Image Token.** Content represents prepending the global content feature, while Semantic denotes prepending the global semantic feature.

**Subject Segmentation**. To mitigate the influence of irrelevant background noise and focus on the primary subject, we leverage the Qwen2.5-VL vision language model (Bai et al., 2025) and the GroundingDINO segmentation model (Liu et al., 2024) to segment the subject from the reference image. We conducted an ablation study, detailed in Table 10, to validate the efficacy of the echo segmentation protocol. The results confirm that the introduction of subject segmentation significantly enhances the generation performance, which is observed in the preservation of subject features, and demonstrate that isolating the main subject is critical to producing more accurate and faithful outputs.

| Exp. | DINO↑ | CLIP-I↑ | CLIP-T↑ |
|---|---|---|---|
| w/o SS | 0.663 | 0.796 | 0.321 |
| w/ SS | **0.672** | **0.806** | 0.321 |

Table 10: **Enhancement by subject segmentation (denoted by SS) to mitigate background noise**.

To further analyze the sensitivity of our method to the quality of segmentation, we conduct an ablation study on about EchoGen's robustness to segmentation quality during inference. Specifically, to simulate segmentation imperfections, we design three variants to simulate disturbances and compare with employing subject segmentation without imperfection during inference: 1. Enlarging Bounding Box: enlarging the subject's bounding box by 10%; 2.

| Exp. | DINO↑ | CLIP-I↑ | CLIP-T↑ |
|---|---|---|---|
| w/o SS | 0.737 | 0.829 | 0.324 |
| Shift | 0.739 | 0.833 | 0.321 |
| Enlarge | 0.735 | 0.831 | 0.321 |
| w/ SS | **0.755** | **0.835** | **0.325** |

Table 11: **Analysis of the sensitivity of EchoGen to the segmentation quality.** SS denotes the subject segmentation; Enlarge and Shift denote enlarging and shifting bounding box, respectively.

Shifting Bounding Box: shifting the bounding box by 10%; 3. No Segmentation: completely removing the subject segmentation step. As shown in Table 11, our model exhibits remarkable robustness as its performance degrades only slightly under these disturbances. Moreover, our model still produces strong results even without any segmentation, demonstrating its powerful generalization capability. In summary, EchoGen is highly robust to imperfect segmentation.

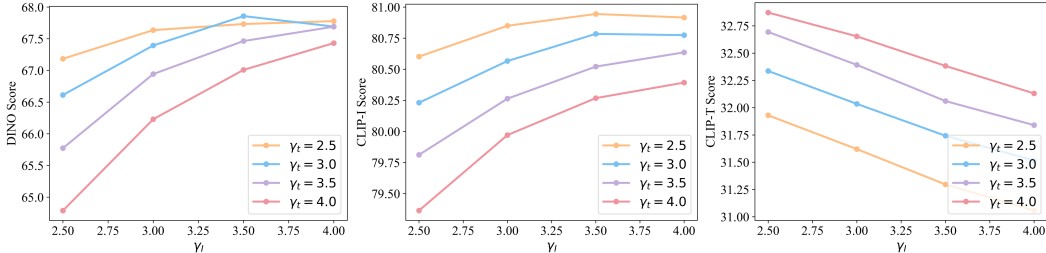

Figure 7: **Visualization of the effect of classifier-free guidance scale coefficient**.

| Method | Base Model | DINO↑ | CLIP-I↑ | CLIP-T↑ | Latency↓ |
|---|---|---|---|---|---|
| IP-Adapter (Ye et al., 2023) | SDXL | 0.713 | 0.851 | 0.293 | 16.9s |
| OminiControl (Tan et al., 2025) | FLUX.1-dev | 0.644 | 0.800 | 0.323 | 27.5s |
| EasyControl (Zhang et al., 2025) | FLUX.1-dev | 0.666 | 0.804 | 0.330 | 25.4s |
| EchoGen-2B | Infinity-2B | 0.755 | 0.835 | 0.325 | 5.2s |

(a) Multi-reference scoring (compare against all references of the same entity).

| Method | Base Model | DINO↑ | CLIP-I↑ | CLIP-T↑ | Latency↓ |
|---|---|---|---|---|---|
| IP-Adapter (Ye et al., 2023) | SDXL | 0.626 | 0.807 | 0.295 | 16.9s |
| OminiControl (Tan et al., 2025) | FLUX.1-dev | 0.575 | 0.771 | 0.323 | 27.5s |
| EasyControl (Zhang et al., 2025) | FLUX.1-dev | 0.600 | 0.773 | 0.330 | 47.6s |
| EchoGen-2B | Infinity-2B | 0.663 | 0.802 | 0.325 | 5.2s |

(b) Single-reference scoring (compare against the corresponding reference image).

Table 12: **Quantitative comparisons on DreamBench (Ruiz et al., 2023) under two evaluation protocols.** Baselines are reproduced from official repositories and evaluated with the same code. EchoGen achieves strong performance with lower latency.

**Subject-text classifier-free guidance**. As detailed in Figure 7, our experiments reveal a clear trade-off governed by the CFG hyperparameters within a proper scope. As the subject guidance weight $\gamma_I$ increases, subject fidelity improves, as indicated by higher CLIP-I and DINO scores. Conversely, this gain is accompanied by reduced text alignment, reflected in lower CLIP-T. The inverse relationship is observed when increasing the text condition scaling coefficient $\gamma_t$. This empirical result demonstrates the efficacy and flexibility of our CFG design, enabling users to dynamically adjust the balance between preserving reference image features and adhering to the text prompt.

**Different evaluation protocols.** We compare our model with several leading baselines under two different evaluation protocols: (i) each generated image is compared against the single corresponding reference image, and (ii) each generated image is compared against all reference images of the same entity in DreamBench. We reproduce the baselines using their official repositories and evaluate all methods with the same evaluation code. As shown in Table 12, our method achieves comparable or superior subject preservation, as measured by DINO and CLIP-I, as well as comparable text alignment (CLIP-T), while maintaining faster sampling speed.

## 7.5 MORE VISUALIZATION RESULTS

We further showcase additional qualitative results on DreamBench in Figure 8. Moreover, we provide additional visual results from the EchoGen-2B model on real-world subject personalization in Figure 9. These results demonstrate that our model, trained exclusively on a filtered combination of the large-scale synthetic Subjects200K (Tan et al., 2025) and UNO-1M (Wu et al., 2025) datasets, our model exhibits strong generalization to real-world scenarios, including the generation of live animals and diverse objects under complex conditions. EchoGen-2B consistently maintains high sub-

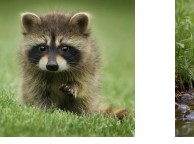
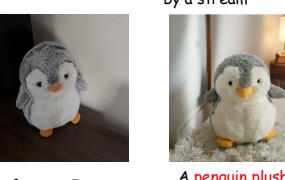
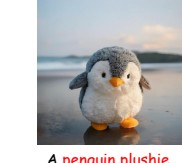

Reference Image · A raccoon is standing by a stream · A raccoon is perched on a tree branch

Reference Image · A penguin plushie on a white rug · A penguin plushie on the beach

Figure 9: **More visualization of EchoGen-2B on real-world subject personalization.**

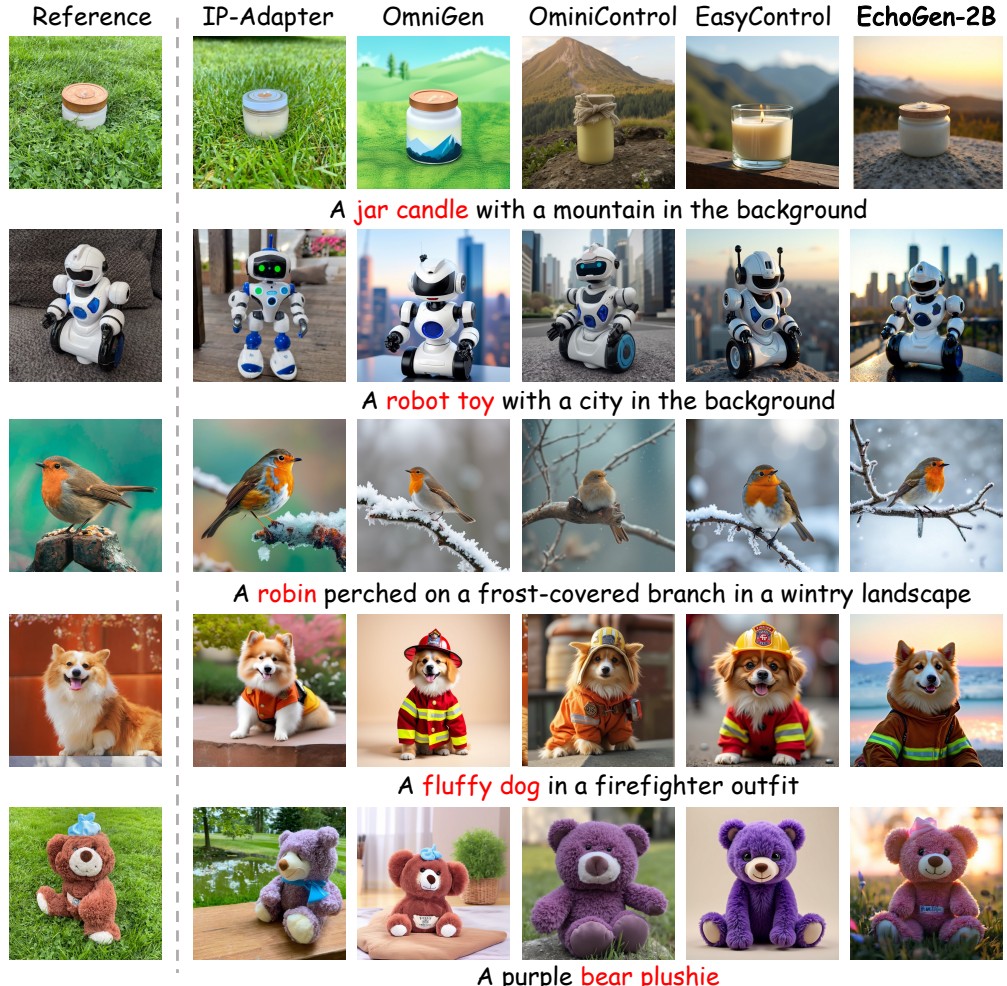

Figure 8: **Additional qualitative comparisons between our EchoGen model and competing methods.**

ject fidelity and strong text alignment during these real-world personalization tasks, demonstrating the effectiveness of our training strategy and the proposed dual-path semantic-content injection design.

## 7.6 LIMITATION & FAILURE CASE ANALYSIS

Our EchoGen takes a new step toward VAR-based feed-forward subject-driven generation to inherit the strong capability of next-scale prediction and bidirectional modeling within scales. However, we know that the feed-forward subject-driven image generation is highly dependent on the capability of base models. The performance of our EchoGen models is fundamentally dependent upon the capability of the base models Infinity-0.1B and Infinity-2B. The Infinity-2B architecture still exhibits a performance gap compared to state-of-the-art generation models such as Stable-Diffusion 3 and FLUX, particularly in generating high-fidelity details. This inherited constraint limits EchoGen's efficacy in resolving fine-grained features, such as the faithful rendering of facial characteristics, the synthesis of coherent text, and the reproduction of intricate material textures. Due to significant GPU computational and temporal constraints, our experiments are confined to these specific backbones, precluding an empirical investigation of larger models such as Infinity-8B. We hypothesize that migrating the EchoGen architecture to a more potent VAR foundation model would unlock substantial performance gains.

Additionally, the DINOv2 vision encoder operates on relatively low-resolution inputs (*e.g.*, 224×224), which limits its ability to capture fine-grained appearance cues and tiny textual elements.

(a) Synthesize complex details                    (b) Render tiny text

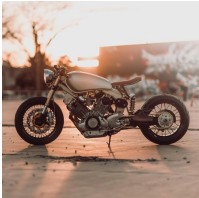 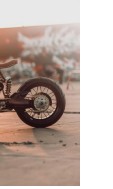    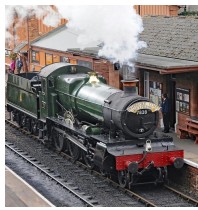 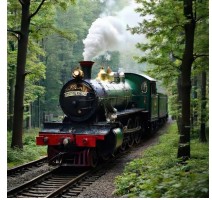

Reference Image    A motorcycle is parked beside a bustling city street at night, illuminated by street lights          Reference Image    A steam locomotive is chugging through a dense forest

Figure 10: **Failure cases generated by EchoGen.**

We believe seeking an effective high-resolution semantic encoder presents a promising avenue for further improvement in complex applications.

Due to the aforementioned limitations, as illustrated in Figure 10, the model exhibits reduced reliability on subjects that have highly intricate structures or in scenarios requiring precise text rendering. We will continue to investigate and address these challenges in future work.

## 7.7 THE USAGE OF LARGE LANGUAGE MODEL

We utilized the large language models Qwen-3 and GPT-5 to improve the clarity, grammar, and formal tone of the writing in the method and experiment sections. Nevertheless, all technical content, such as conceptual formulas and experiments remain our own; the large language models were used solely as tools for linguistic enhancement and stylistic polishing.

