# OpenReview forum: "EchoGen: Generating Visual Echoes in Any Scene via Feed-Forward Subject-Driven Auto-Regressive Model"
_ICLR.cc/2026/Conference — ICLR 2026 Poster_

### Official Review · Reviewer_uSYJ · 2025-10-27

**Soundness:** 3
**Presentation:** 4
**Contribution:** 4
**Rating:** 6
**Confidence:** 3

**Summary:**

The paper introduces EchoGen, a feed-forward subject-driven image generation framework that replaces the diffusion backbone with a visual autoregressive (VAR) architecture. The model integrates a dual-path feature injection mechanism: (1) a semantic path using DINOv2 features to capture high-level identity and structure, and (2) a content path using FLUX.1-dev VAE features to preserve low-level details. This design enables EchoGen to synthesize novel scenes containing a given subject without per-subject fine-tuning. On benchmarks such as DreamBench, EchoGen is competitive with diffusion-based approaches in performance, while offering faster inference.

**Strengths:**

1. This paper presents the first subject-driven feed-forward framework on a VAR backbone with a well-motivated dual-path injection design to capture both the high-level and low-level information from subject images.
2. EchoGen achieves competitive subject-driven image generation performance to state-of-the-art diffusion methods while cutting sampling latency to 5.2 seconds (vs. 10-50 seconds for feed-forward diffusion models).

**Weaknesses:**

1. The efficiency comparison in Table 1 may not be fully fair. The paper reports using default sampling configurations for all diffusion baselines, yet sampling time in diffusion models is highly dependent on sampling parameters such as the number of denoising steps. This can potentially overstate EchoGen’s relative speed advantage. A more rigorous analysis showing performance-compute tradeoffs across multiple sampling settings of the baselines would make the efficiency claims of EchoGen more convincing.

**Questions:**

1. The Subjects-200k dataset appears largely synthetic (generated with GPT-4o and FLUX.1-dev). How well does EchoGen generalize to real-world subject personalization?

---

> ### Author Response · Authors · 2025-11-24
> **Response to Reviewer uSYJ**
>
> **Q1: Performance-latency Trade-offs.**
>
> A1: Thank you for your valuable comments! Indeed, the sampling latency of diffusion-based approaches is highly dependent on the number of denoising steps. We have conducted a more thorough analysis to evaluate the performance-latency trade-offs across all methods. Specifically, we evaluate diffusion-based methods with varying numbers of denoising steps and report their performance versus sampling latency in Figure 5. For diffusion baselines, increasing the number of denoising steps improves subject fidelity (DINO, CLIP-I scores) up to a saturation point. In contrast, the text alignment (CLIP-T) score converges much earlier. **Our model consistently offers a better trade-off**, achieving strong performance with significantly lower sampling latency than the baseline diffusion-based models. This confirms the inherent efficiency and effectiveness of our approach.
>
> **Q2:Generalization to Real-world Subject Personalization.**
>
> A2: We respectfully clarify that constructing a real-world, large-scale, subject-driven dataset with triplets of (reference image, text prompt, generated sample) is prohibitively expensive and time-consuming. Therefore, our approach follows established practices in recent works such as OmniControl[2] and EasyControl[3]. We utilized the synthetic Subjects200K dataset for training, then rigorously evaluated EchoGen on the DreamBench benchmark, which consists of **real-world** images with prompt annotations. As demonstrated in the Experiment Section, our model exhibits excellent generalization capabilities in real-world applications. Additionally, we provide more visual results from the EchoGen-2B model on real-world subject personalization in Figure 9. These results demonstrate that, with training exclusively on the large-scale synthetic dataset Subjects200K, it **exhibits strong generalization to real-world scenarios**, including the generation of live animals and various objects under complex conditions. EchoGen-2B consistently preserves high subject fidelity and strong text alignment during real-world personalization tasks, demonstrating the effectiveness of our training strategy and the proposed dual-path semantic-content incorporation design.
>
> **References:**
>
> [1] Ominicontrol: Minimal and universal control for diffusion transformer.
>
> [2] Easycontrol: Adding efficient and flexible control for diffusion transformer.

---

### Official Review · Reviewer_6ZGd · 2025-10-29

**Soundness:** 3
**Presentation:** 3
**Contribution:** 3
**Rating:** 4
**Confidence:** 4

**Summary:**

EchoGen is proposed as the first feed-forward, subject-driven image generation framework based on Visual Auto-Regressive (VAR) models, overcoming the limitations of existing methods that trade off controllability for speed. By employing a dual-path injection mechanism—decoupling high-level semantics (via DINOv2) and low-level details (via FLUX.1-dev VAE) through specialized attention modules—EchoGen enables fast, non-iterative sampling. Efficiently trained on Subjects-200k and evaluated on DreamBench, EchoGen achieves subject fidelity and text alignment on par with or better than leading diffusion models, while reducing inference time for 1024×1024 images to under 6 seconds.

**Strengths:**

1. The separation of semantic identity and content details is technically sound. Leveraging DINOv2 for semantic abstraction and FLUX VAE for detail encoding, complemented by distinct attention mechanisms, demonstrates appropriate feature representation in subject-driven image generation.
2. EchoGen demonstrates comparable or improved results relative to strong diffusion-based feed-forward methods (e.g., IP-Adapter, OminiControl) on DreamBench benchmarks (CLIP-I, DINO, CLIP-T). Its superiority in subject fidelity and photorealism is further corroborated by human evaluation.
3. A latency of under 6 seconds for high-resolution image generation represents a significant practical advantage, making EchoGen particularly well-suited for real-time and user-facing applications where responsiveness is critical.

**Weaknesses:**

1. The authors mention related work [1] on fine-tuning VAR for subject-driven generation, but this is not included in experiments. A comparison would help clarify whether feed-forward is truly superior to lightweight fine-tuning in the VAR regime.
2. Although Subjects-200k (~256k triplets) is substantial in scale, it is synthetically generated using FLUX.1-dev and GPT-4o, which may introduce distributional biases and limit real-world subject diversity. Furthermore, the paper does not provide a detailed analysis of failure cases involving challenging real-world subjects, such as humans in complex poses or transparent objects.
3. While Table 2 reports preference ratios, important methodological details are missing, such as the evaluation format (single-choice vs. multi-select), the number of images per trial, and participant expertise. Including this information would strengthen the reliability of the human study findings.

[1] Fine-tuning visual autoregressive models for subject-driven generation

**Questions:**

Please refer to the weaknesses. If the issues I raised are addressed, I will increase my rating.

---

> ### Author Response · Authors · 2025-11-24
> **Response to Reviewer 6ZGd**
>
> **Q1: Comparison with AR-Booth.**
>
> A1: We thank the reviewer for this valuable suggestion. Since AR-Booth[1] did not report their results on DreamBench, we have implemented their method ourselves, updated the results in Table 1, and also reported them in Table R3.1 for direct comparison. We observe that EchoGen achieves higher DINO and CLIP-I scores, indicating **better subject fidelity**. More importantly, EchoGen obtains a significantly higher CLIP-T score, which demonstrates its **superior text alignment capabilities** compared to the test-time fine-tuning approach. The superior performance of our method can be attributed to our dual-path semantic-content integration strategy, which ensures more effective information fusion. Instead, test-time fine-tuning methods like AR-Booth require costly per-subject fine-tuning, leading to substantially higher generation latency when encountering an unseen subject. In contrast, EchoGen's feed-forward nature eliminates this bottleneck, making it **significantly more efficient and practical** for real-world applications.
>
> > **Table R3.1:** Comparison with the recent test-time fine-tuning AR-Booth method based on VAR.
>
> | Method                          | DINO↑   | CLIP-I↑ | CLIP-T↑ | Latency↓ |
> |:-------------------------------:|:-------:|:-------:|:-------:|:--------:|
> | AR-Booth[1]                    | 0.750   | 0.808   | 0.269   | 2.8h     |
> | EchoGen-2B                      | **0.755**   | **0.837**   | **0.324**   | **5.2s**     |
>
> **Q2: Usage of the Synthetic Dataset.**
>
> A2: We respectfully clarify that constructing a real-world, large-scale, subject-driven dataset with triplets of (reference image, text prompt, generated sample) is prohibitively expensive and time-consuming. Therefore, our approach follows established practices in recent works such as OmniControl[2] and EasyControl[3]. We utilized the synthetic Subjects200K dataset for training, then rigorously evaluated EchoGen on the DreamBench benchmark, which consists of **real-world** images with prompt annotations. As demonstrated in the Experiment Section, our model exhibits excellent generalization capabilities in real-world applications. Additionally, we provide more visual results from the EchoGen-2B model on real-world subject personalization in Figure 9. These results demonstrate that, with training exclusively on the large-scale synthetic dataset Subjects200K, it **exhibits strong generalization to real-world scenarios**, including the generation of live animals and various objects under complex conditions. EchoGen-2B consistently preserves high subject fidelity and strong text alignment during real-world personalization tasks, demonstrating the effectiveness of our training strategy and the proposed dual-path semantic-content incorporation design.
>
> **Q3: Analyzing Failure Cases.**
>
> A3: We acknowledge that EchoGen has some limitations. As illustrated in Figure 10, the model performs less reliably on subjects with **highly intricate structures** or on cases requiring **precise text rendering**. We attribute these shortcomings primarily to *the capacity constraints of both the Infinity-2B backbone and the semantic encoder*. Specifically, the limited expressive power and insufficient prior knowledge of the Infinity-2B architecture restrict the generation of high-fidelity details. Additionally, the DINOv2 vision encoder operates on relatively low-resolution inputs (*e.g.*, 224×224), which impairs its ability to capture fine-grained appearance cues and tiny text elements. We believe scaling the Infinity-2B backbone and seeking an effective high-resolution semantic encoder is a promising direction for further improvement in complex applications.
>
> **Q4: Human Evaluation Details.**
>
> A4: We thank the reviewer for this question. For human evaluation, we conducted a comprehensive study comparing EchoGen with **four leading methods**: IP-Adapter[4], OmniGen[5], OminiControl[2], and EasyControl[3]. For each method, we generated 18 samples given the same prompts, without any cherry-picking. Then, the evaluation employed a **multi-selection mechanism** -- participants were asked to choose the best from the provided five generated samples under three distinct criteria: Subject Fidelity, Text Alignment, and Photorealism. We collected responses from 25 participants, all of them with expertise in generative models. The preference ratios for each metric are reported in Table 2, which demonstrates a clear user preference for EchoGen.
>
> **References:**
>
> [1] Fine-tuning visual autoregressive models for subject-driven generation.
>
> [2] Ominicontrol: Minimal and universal control for diffusion transformer.
>
> [3] Easycontrol: Adding efficient and flexible control for diffusion transformer.
>
> [4] Ip-adapter: Text compatible image prompt adapter for text-to-image diffusion models.
>
> [5] Omnigen: Unified image generation.

---

> ### Comment · Reviewer_6ZGd · 2025-11-28
> **Comment**
>
> Thanks for your reply. The main concerns of mine have been addressed, and I will increase the rate to 6 as a positive score.

---

> > ### Author Response · Authors · 2025-12-03
> > **Official Comment by Authors**
> >
> > We are truly grateful for your decision to raise the score. Thank you again for your valuable time and insightful guidance.

---

### Official Review · Reviewer_aSDF · 2025-11-01

**Soundness:** 3
**Presentation:** 3
**Contribution:** 3
**Rating:** 6
**Confidence:** 4

**Summary:**

This paper introduces EchoGen, the first subject-driven image generation framework built on a VAR model rather than diffusion models. The method enables fast, feed-forward subject-driven generation by injecting both high-level semantic identity and low-level visual details through a dual-path design. EchoGen leverages a semantic encoder and a content encoder to disentangle identity and appearance, achieving controllable and high-fidelity results. Experiments show that EchoGen reaches state-of-the-art subject fidelity and image quality while significantly reducing sampling latency compared to diffusion-based approaches.

**Strengths:**

1. The paper presents the first subject-driven image generation framework based on a visual autoregressive model, expanding the design space beyond the diffusion-based paradigm.
2. The dual-path injection strategy—separating semantic identity from fine-grained visual content—is conceptually meaningful and empirically shown to improve subject fidelity and controllability.
3. The method achieves competitive or superior generation quality compared to diffusion-based baselines while offering substantially lower inference latency, addressing a clear practical limitation of existing approaches.

**Weaknesses:**

1. Although the framework claims to disentangle semantic and content features, the paper does not provide explicit mechanisms or empirical evidence demonstrating effective disentanglement between these two feature types.
2. The method depends on multiple large pretrained models (Qwen2.5-VL, DINOv2, FLUX.1-dev VAE), which increases system complexity and may limit applicability in real-world or resource-constrained settings.
3. The motivation for adopting VAR models feels somewhat shallow—mainly framed as “VAR is faster than diffusion”—without deeper justification for why VAR is inherently suitable for subject-driven generation beyond latency.
4. Some ablation studies and qualitative analyses are missing, such as the effect of removing DINOv2 or FLUX.1-dev, or visual comparisons showing the contribution of semantic vs. content features.

**Questions:**

1. The pipeline relies on several large pretrained components (Qwen2.5-VL, DINOv2, FLUX.1-dev VAE, and the base VAR model). Do the latency numbers reported in the paper reflect the full end-to-end runtime, including all feature extraction modules, or only the VAR sampling stage?
2. Is there a comparison of computational resource usage (e.g., GPU memory, inference throughput) against diffusion-based subject-driven methods? Such information would clarify the practical efficiency advantages claimed in the paper.
3. Why is the semantic feature injected into the Image Token instead of the content feature? Is there a conceptual or empirical advantage to this setting, and were alternative designs explored?

---

> ### Author Response · Authors · 2025-11-24
> **Response to Reviewer aSDF (1/2)**
>
> **Q1: Empirical Evidence of Disentangling Semantic and Content Features.**
>
> A1: Thank you for this question. Our design choice to treat DINOv2 as a semantic encoder and FLUX.1-dev VAE as a content encoder is based on their respective **training objectives** and also **validated by extensive prior works**[1,2]. DINOv2, as a vision foundation model trained for discriminative learning, produces features rich in abstract, high-level information. Conversely, the FLUX.1-dev VAE is trained for reconstruction, which causes it to focus on low-level content details such as texture and color. This distinction is supported by previous works (*e.g.*, VA-VAE[3], REPA[4], RAE[5]) that often uses semantic alignment losses or leverages a semantic pre-trained encoder to enhance VAE representations.
>
> *The Experiment Section confirms this division of labor.* Table 4 shows that DINOv2 features guide the global structure and fine-grained fidelity of the subject, while simply injecting detailed information fails due to its lack of semantics. Table 7 further demonstrates that the introduction of content features improves detail quality and subject fidelity, acting as a complementary source of information. Beyond quantitative comparisons, we provide a qualitative analysis of the effect of the semantic and content feature injection in Figure 6. We observe that with semantic injection alone, our model yields a significant improvement in fidelity, and with the addition of content feature injection, the local details get further refined. This dual-injection strategy is the key to the performance of our model.
>
> **Q2: Sampling Latency.**
>
> A2: We apologize for the ambiguity regarding the sampling latency. Table R2.1 provides a detailed latency breakdown of each component, demonstrating that our framework's efficiency is *not impacted by Qwen2.5-VL and GroundingDINO*, and that it remains **significantly faster** than diffusion-based methods even leveraging Qwen2.5-VL. We would like to emphasize that Qwen2.5-VL is not a prerequisite for inference: it serves as a tool to automatically identify the primary subject of the reference image and provide a descriptive phrase for the Grounding-DINO segmentation model in the training stage, mainly because our training dataset lacks explicit annotations about the region of the primary subject. During inference, the framework does not rely on Qwen2.5-VL. The user can simply provide a text prompt that describes the subject, similar to the annotations provided in the DreamBench benchmark. Moreover, according to Table R2.1, the computational cost of the Grounding-DINO model and the semantic/content image encoder is negligible compared to the overall sampling latency during inference.
>
> > **Table R2.1:** Per-component sampling latency measured on a single H20 GPU.
>
> | Model Component           | Sampling Latency (s) |
> |:---------:|:---------:|
> | Grounding-DINO            | 0.24                 |
> | DINOv2                    | 0.008                |
> | FLUX.1-dev VAE            | 0.025                |
> | Infinity Generator        | 4.95                 |
> | Qwen2.5-VL (Optional)     | 1.13                 |
> | EchoGen (w/o Qwen2.5-VL)  | 5.22                 |
> | EchoGen (w/ Qwen2.5-VL)   | 6.35                 |
>
> **Q3: Motivation for Employing the Visual Autoregressive (VAR) Framework.**
>
> A3: We thank the reviewer for this insightful point. We adopt the VAR framework[6] due to its compelling combination of **superior performance** and **efficiency**. In text-to-image generation, recent VAR models such as Infinity[7] have shown results competitive with top-tier diffusion models (*e.g.*, SD3[8], FLUX[9]), but with substantially lower sampling latency. This impressive performance-efficiency profile has led to the successful application of VAR in areas such as large-scale foundation models[10] and controllable generation[11,12]. Despite these successes in related applications, feed-forward, subject-driven generation with this promising efficient framework still remains underexplored. In this paper, we carefully design an effective dual-path injection mechanism that enables feed-forward subject-driven image generation using VAR.

---

> ### Author Response · Authors · 2025-11-24
> **Response to Reviewer aSDF (2/2)**
>
> **Q4: Qualitative Analysis of the Effect of the Semantic and Content Feature Injection.**
>
> A4: Thanks for pointing this out. As illustrated in Figure 6, we qualitatively dissect the effect of each feature component. Starting from the base Infinity backbone, we observe that introducing semantic features extracted by DINOv2 enables the generator to synthesize subjects faithful to the reference subject's structure and style. When further incorporating content features from the FLUX.1-dev VAE, our EchoGen demonstrates significantly improved capability to render fine-grained, coherent details (*e.g.*, the facial features of the robot toy and the fluffy dog, as well as the material and color of the shoe uppers).
>
> **Q5: Comparison of Computation Resource Usage.**
>
> A5: Good question! In Table 1, we have shown a direct comparison of efficiency via inference throughput with leading diffusion-based methods, including IP-Adapter[13], OmniGen[14], OminiControl[15], and EasyControl[16]. The results demonstrate that EchoGen not only achieves performance superior to these methods in terms of subject fidelity and text alignment, *but also exhibits significantly higher inference throughput and lower sampling latency*.
>
> **Q6: Importance of Injecting Global Semantic Information.**
>
> A6: Injecting global semantic information serves as a prepended condition, to ensure global structural coherence during generation. Our ablation study in Table 5 confirms the significant benefits of incorporating global semantic features. Moreover, we conducted a targeted experiment comparing the injection of global semantic features versus content features to the Image Token. As shown in Table R2.2, the results clearly indicate that *prepending semantic features rather than content information into the image token significantly enhances the subject fidelity*. This confirms that our choice to inject global semantic guidance into image tokens is both effective and well-justified.
>
> > **Table R2.2:** Analysis of different information types prepended to the Image Token. Content represents prepending the global content feature, while Semantic denotes prepending the global semantic feature.
>
> | Exp.      | DINO↑     | CLIP-I↑   | CLIP-T↑     |
> |:---------:|:---------:|:---------:|:-----------:|
> | Content   | 0.663     | 0.795     | **0.322**   |
> | Semantic  | **0.672** | **0.806** | 0.321       |
>
> **References:**
>
> [1] Imagefolder: Autoregressive image generation with folded tokens.
>
> [2] Latent diffusion model without variational autoencoder.
>
> [3] Taming optimization dilemma in latent diffusion models.
>
> [4] Representation alignment for generation: Training diffusion transformers is easier than you think.
>
> [5] Diffusion transformers with representation autoencoders.
>
> [6] Visual autoregressive modeling: Scalable image generation via next-scale prediction.
>
> [7] Infinity: Scaling bitwise autoregressive modeling for high-resolution image synthesis.
>
> [8] Infinitystar: Unified spacetime autoregressive modeling for visual generation.
>
> [9] Car: Controllable autoregressive modeling for visual generation.
>
> [10] Controlvar: Exploring controllable visual autoregressive modeling.
>
> [11] Scaling rectified flow transformers for high-resolution image synthesis
>
> [12] Flux. 1 kontext: Flow matching for in-context image generation and editing in latent space.
>
> [13] Ip-adapter: Text compatible image prompt adapter for text-to-image diffusion models.
>
> [14] Omnigen: Unified image generation.
>
> [15] Ominicontrol: Minimal and universal control for diffusion transformer.
>
> [16] Easycontrol: Adding efficient and flexible control for diffusion transformer.

---

> ### Comment · Reviewer_aSDF · 2025-11-25
>
> Thanks to the authors for providing additional results and clarifications. Most of my concerns have now been addressed. Since I previously gave a positive score, I will keep my rating.

---

> > ### Author Response · Authors · 2025-12-03
> > **Official Comment by Authors**
> >
> > We sincerely thank you for your follow-up and for acknowledging that our clarifications addressed your concerns. We appreciate your positive assessment and your decision to maintain the current positive rating.

---

### Official Review · Reviewer_YBoY · 2025-11-01

**Soundness:** 4
**Presentation:** 4
**Contribution:** 3
**Rating:** 6
**Confidence:** 4

**Summary:**

This paper introduces EchoGen, a novel feed-forward framework for subject-driven image generation built upon a Visual Autoregressive (VAR) model. Motivated by the significant latency of diffusion-based approaches and the high computational cost of per-subject fine-tuning , EchoGen leverages the inherent fast sampling of VAR models to address this efficiency gap. The core technical contribution is a dual-path injection strategy that disentangles a subject's high-level semantic identity from its fine-grained textural details, enabling high-fidelity preservation and control. Experimental results validate that EchoGen achieves subject fidelity and text alignment competitive with state-of-the-art diffusion-based methods, while being significantly faster at inference.

**Strengths:**

The paper’s primary strength lies in its novel and successful exploration of an autoregressive architecture for subject-driven generation. This approach achieves performance that is highly competitive with state-of-the-art diffusion-based methods. Its most notable advantage is that it delivers strong results while drastically reducing inference latency, offering a significant improvement in efficiency and practicality compared to the slow, iterative sampling required by diffusion models. Moreover, the empirical evaluation is thorough and rigorous, supporting the validity of the proposed approach.

**Weaknesses:**

1. The model's generative quality is fundamentally limited by the capabilities of its Infinity-2B backbone. This backbone restricts its ability to render fine-grained details like coherent text or complex textures, meaning its peak quality is constrained even though its efficiency is high.
2. The method relies on a complex pre-processing pipeline using Qwen2.5-VL and GroundingDINO, which adds computational overhead. It is unclear if the reported inference latencies (5.2s for EchoGen-2B ) include this segmentation step. If not, the true "wall-clock" efficiency comparison is ambiguous.
3. The paper lacks an analysis of the model's sensitivity to the quality of the segmentation. While an ablation confirms segmentation is beneficial, it is unknown how performance would be impacted by a less accurate (or more accurate) segmentation model or by common bounding box errors. The system's robustness to imperfect pre-processing remains untested.

**Questions:**

See the weakness section above.

---

> ### Author Response · Authors · 2025-11-24
> **Response to Reviewer YBoY**
>
> **Q1: Regarding the Infinity-2B Backbone.**
>
> A1: We thank the reviewer for this insightful comment. We employed the Infinity-0.1B and Infinity-2B models as our backbone, primarily due to significant GPU resource constraints. We acknowledge that this may limit the generation of the highest-fidelity details, but our comprehensive and fair experiments provide strong evidence of the effectiveness and **scalability** of our EchoGen approach. The performance gains observed when scaling EchoGen-0.1B to EchoGen-2B suggest that integrating our method with larger backbones would unlock substantial improvements in generation quality. Motivated by this trend, *we are seeking more GPU resources to further implement our EchoGen-8B model based on the Infinity-8B model.*
>
> **Q2: Sampling Latency.**
>
> A2: We apologize for the ambiguity regarding the sampling latency. Table R1.1 provides a detailed latency breakdown of each component, demonstrating that our framework's efficiency is *not impacted by Qwen2.5-VL and GroundingDINO*, and that it remains **significantly faster** than diffusion-based methods even leveraging Qwen2.5-VL. We would like to emphasize that Qwen2.5-VL is not a prerequisite for inference: it serves as a tool to automatically identify the primary subject of the reference image and provide a descriptive phrase for the Grounding-DINO segmentation model in the training stage, mainly because our training dataset lacks explicit annotations about the region of the primary subject. During inference, the framework does not rely on Qwen2.5-VL. The user can simply provide a text prompt that describes the subject, similar to the annotations provided in the DreamBench benchmark. Moreover, according to Table R1.1, the computational cost of the Grounding-DINO model is negligible compared to the overall sampling latency during inference.
>
> > **Table R1.1:** Per-component sampling latency measured on a single H20 GPU.
>
> | Model Component           | Sampling Latency (s) |
> |:---------:|:---------:|
> | Grounding-DINO            | 0.24                 |
> | DINOv2                    | 0.008                |
> | FLUX.1-dev VAE            | 0.025                |
> | Infinity Generator        | 4.95                 |
> | Qwen2.5-VL (Optional)     | 1.13                 |
> | EchoGen (w/o Qwen2.5-VL)  | 5.22                 |
> | EchoGen (w/ Qwen2.5-VL)   | 6.35                 |
>
> **Q3: Robustness to Segmentation Quality.**
>
> A3: Thanks for this insightful comment! We conducted an ablation study about EchoGen's robustness to segmentation quality during inference. We note that while the subject segmentation mitigates the influence of irrelevant background noise, EchoGen exhibits systematic robustness and generalization with imperfect segmentation. Specifically, to simulate segmentation imperfections, we designed three variants to simulate disturbances and compared with employing subject segmentation without imperfection during inference.
>
> - Enlarging Bounding Box: enlarging the subject's bounding box by 10%;
>
> - Shifting Bounding Box: shifting the bounding box by 10%;
>
> - No Segmentation: completely removing the subject segmentation step.
>
> As shown in Table R1.2, our model exhibits remarkable robustness as its performance degrades only slightly under these disturbances. Moreover, our model still produces strong results even without any segmentation, demonstrating its powerful generalization capabilities. In summary, EchoGen is **highly robust to imperfect segmentation**.
>
> > **Table R1.2:** Analysis of the sensitivity of our method to the quality of the segmentation. SS denotes subject segmentation; Enlarge and Shift denote enlarging and shifting the bounding box, respectively.
>
> | Exp.               | DINO↑ | CLIP-I↑ | CLIP-T↑ |
> |:------------------:|:-----:|:-------:|:-------:|
> | w/o SS             | 0.737 |  0.829  |  0.324  |
> | Shift              | 0.739 |  0.833  |  0.321  |
> | Enlarge            | 0.735 |  0.831  |  0.321  |
> | w/ SS              | **0.755** |  **0.837**  |  0.324  |

---

> > ### Comment · Reviewer_YBoY · 2025-11-27
> >
> > I thank the authors for their clarifications. My concerns have now been addressed. Given that I initially assigned a positive score, I will maintain my current rating.

---

> > > ### Author Response · Authors · 2025-12-03
> > > **Official Comment by Authors**
> > >
> > > Thank you for your time and the constructive comments. We appreciate your response and are pleased that our clarifications have addressed your concerns.

---

### Author Response · Authors · 2025-11-24
**General Response**

We thank the reviewers for their valuable and insightful feedback. Our paper introduces **EchoGen**, **the first feed-forward subject-driven visual autoregressive framework** employing a dual semantic-content injection strategy to enhance subject fidelity while preserving text alignment. EchoGen outperforms diffusion-based methods in both benchmark comparisons and human evaluations, while significantly reducing sampling latency to 5.2 seconds (vs.>10s for baselines).

We have responded to all the questions raised by the reviewers and carefully revised the manuscript based on the constructive suggestions. If there are any additional questions, we will be happy to engage in further discussion. The modifications of the manuscript are highlighted in blue.

---

### Meta-Review · Area_Chair_7nLD · 2026-01-11

**Summary:**

Most reviews (except one) were initially positive, raising a number of concerns about quality limitations, evidence of disentanglement, missing ablations, and more---but any and all concerns seem to have been addressed in the author rebuttals.

**Reviewer Concerns:**

The concerns that were raised were:

- The fact that the model's quality is limited by the backbone
- The fact that the inference latency doesn't account for the time spend preprocessing
- Missing analysis about how the errors in bounding box/segmentation can affect results
- Lack of evidence for disentanglement of semantics and details.
- The system complexity.
- Unclear motivation for the use of visual autoregressive models.
- Missing ablations (removing DINO/Flux).
- Limitations from training on synthetic data---issues with generalization to real-world
- Missing methodology for human evals.
- Unfair comparison with diffusion models

The rebuttal seems to have addressed ALL of these concerns, and most reviewers have responded acknowledging this and adjusting their scores

**Reviewer Scores:**

Scores were mostly positive, with one exception. The lower score has mentioned in a response that they intend to raise their score, and although it does not seem to have come through, it should be expected that all scores are currently positive (6+).

---

### Decision · Program_Chairs · 2026-01-26

Accept (Poster)